# Stringent Response Factor DksA Contributes to Fatty Acid Degradation Function to Influence Cell Membrane Stability and Polymyxin B Resistance of *Yersinia enterocolitica*

**DOI:** 10.3390/ijms241511951

**Published:** 2023-07-26

**Authors:** Can Huang, Wenqian Li, Jingyu Chen

**Affiliations:** Beijing Laboratory for Food Quality and Safety, College of Food Science & Nutritional Engineering, China Agricultural University, 17 Qinghua East Rd., Beijing 100083, China

**Keywords:** *Yersinia enterocolitica*, DksA, fatty acid metabolism, polymyxin B

## Abstract

DksA is a proteobacterial regulator that binds directly to the secondary channel of RNA polymerase with (p)ppGpp and is responsible for various bacterial physiological activities. While (p)ppGpp is known to be involved in the regulation and response of fatty acid metabolism pathways in many foodborne pathogens, the role of DksA in this process has yet to be clarified. This study aimed to characterize the function of DksA on fatty acid metabolism and cell membrane structure in *Yersinia enterocolitica*. Therefore, comparison analysis of gene expression, growth conditions, and membrane permeabilization among the wide-type (WT), DksA-deficient mutant (YEND), and the complemented strain was carried out. It confirmed that deletion of DksA led to a more than four-fold decrease in the expression of fatty acid degradation genes, including *fadADEIJ*. Additionally, YEND exhibited a smaller growth gap compared to the WT strain at low temperatures, indicating that DksA is not required for the growth of *Y. enterocolitica* in cold environments. Given that polymyxin B is a cationic antimicrobial peptide that targets the cell membrane, the roles of DksA under polymyxin B exposure were also characterized. It was found that DksA positively regulates the integrity of the inner and outer membranes of *Y. enterocolitica* under polymyxin B, preventing the leakage of intracellular nucleic acids and proteins and ultimately reducing the sensitivity of *Y. enterocolitica* to polymyxin B. Taken together, this study provides insights into the functions of DksA and paves the way for novel fungicide development.

## 1. Introduction

*Yersinia enterocolitica* is a common foodborne pathogen which causes a range of acute gastrointestinal illnesses known as yersiniosis in humans. The symptoms include vomiting, abdominal pain, and diarrhea, with complications such as mesenteric lymphadenitis, reactive arthritis, and sepsis in severe cases [1,2]. According to the data from the US Centers for Disease Control and Prevention (CDC), there are approximately 117,000 cases of yersiniosis in the United States each year, with 90% of cases resulting from the consumption of contaminated food [3]. In Europe, yersiniosis ranks as the third most prevalent foodborne zoonotic disease after campylobacteriosis and salmonellosis [4]. *Y. enterocolitica* is widely distributed in the environment and has been detected in various animals such as pigs, cattle, and dogs, as well as in rivers, soils, and prepared environments. It can also be found in foods such as pork, dairy, soybean, and aquatic products. The living conditions in these environments are extremely complex, and the pathogen may encounter diverse nutritional and environmental stresses, including acid, alkali, low temperature, hyperosmotic pressure, and antimicrobial peptides [5]. To survive in these challenging conditions, *Y. enterocolitica* have evolved various feasible and efficient environmental response systems, such as stringent response [6] and the two-component system [7]. These systems help the bacteria adjust their metabolism networks and synthesize cytokines, thereby enhancing their chances of survival. Understanding the stress-resistance mechanisms of *Y. enterocolitica* is crucial for developing novel control strategies for this pathogen in the food industry.

The stringent response is an environmental response system that utilizes “magic spot” guanosine-5′, 3′-tetraphosphate ppGpp and guanosine-5′, and 3′-pentaphosphate pppGpp (collectively referred to as (p)ppGpp) as the secondary messengers in bacteria [8]. During exponential growth, (p)ppGpp is present at basal levels and helps regulate the bacterial growth rate and metabolism [9]. However, when the external environment is deficient in nutrition, the signal triggers an increase in cellule (p)ppGpp concentration and regulates hundreds of downstream targeted genes involved in various physiological processes, such as bacterial amino acid transport and utilization, virulence, and environmental tolerance [10,11]. The stringent response was first observed to be induced by amino acid starvation, resulting in a severe decline in rRNA and tRNA synthesis [12]. Subsequent studies have confirmed that this process is primarily caused by changing RNAP activity and has a wide range of effects on bacterial physiological processes in vivo [13,14,15]. However, the effect of (p)ppGpp on transcription was not obvious in vitro. This contradictory result suggests that other regulatory factors may be involved in the regulation of the stringent response.

DksA is a transcriptional regulatory factor composed of 151 amino acids, which was originally identified as a multicopy suppressor of the DnaK mutant [16]. It was later proposed that DksA plays a role in translation and was found to bind RNAP at its basal concentration [17]. In the simulated transcription system in vitro, the addition of DksA will significantly reduce the inhibition level of (p)ppGpp on the rRNA promoter from one-third of the original expression level to one-twentieth [18]. Furthermore, DksA-knockout strains exhibited similar phenotypes to (p)ppGpp-knockout strains, with both showing lower growth rates and virulence, as well as very similar transcriptional profiles [19,20,21]. These studies collectively indicate that DksA is an important regulator of stringent responses and is closely related to (p)ppGpp function. It is now known that (p)ppGpp and DksA dock on the secondary channel of the β’ subunit, and the aspartic acid residue in the helical structure approaches the RNAP active site, strengthening the signaling of (p)ppGpp during the translation [22,23]. Transcriptome results have shown that DksA regulates more than 7% of genes in *Escherichia coli* [24] and over 20% of the genes in *Xanthomonas citri* [25]. These genes are involved in various physiological processes, such as the type 3 secretion system (T3SS), motility, biofilm formation, ribosome synthesis, and others.

Fatty acid degradation is a fundamental metabolism in bacteria, which not only generates a large amount of energy and materials for the cell, but also maintains the homeostasis of cell membrane lipids in growth and stationary phases [26]. Several studies have reported the association of (p)ppGpp with bacterial fatty acid metabolism. For example, fatty acid starvation is an important trigger of stringent responses. During fatty acid starvation, the acyl carrier protein (ACP) is not charged, and the uncharged ACP binds to SpoT, a member of the bifunctional RelA-SpoT homologue family enzymes in most beta- and gamma-proteobacteria, to induce the synthesis of (p)ppGpp [27]. Furthermore, it has been reported that YtfK interacts with the N-terminal region of SpoT to trigger a stringent response [28]. (p)ppGpp also has a significant regulatory effect on fatty acid metabolism pathways. Matthew et al. reported that lack of (p)ppGpp would cause significant changes in the expression levels of 17 fatty acid synthesis genes and 6 fatty acid β-oxidation genes [29]. In addition, (p)ppGpp can bind and inhibit PlsB and PgsA, which are responsible for lipid and phospholipid biosynthesis, leading to the inhibition of the carboxyltransferase complex activities [30,31]. These studies collectively indicate that (p)ppGpp affects fatty acid metabolism in multiple ways, but the effect of DksA on fatty acid metabolism remains unclear. This study aims to characterize the functions of DksA in the fatty acid metabolism and cell membrane integrity of *Y. enterocolitica*. Additionally, the effect of DksA on the bactericidal mechanism of polymyxin B in *Y. enterocolitica* was identified. Results from this study will provide insights into the regulatory role of DksA in the cell membrane structure in *Y. enterocolitica* and assist in identifying a valuable target for controlling *Y. enterocolitica* in food.

## 2. Results

### 2.1. DksA Positively Regulated the Expression Level of Genes Related to the Fatty Acid Degradation Pathway in Y. enterocolitica

Bacteria degrade intracellular fatty acids through the β-oxidation pathway, which not only produces a substantial amount of energy and materials for cell metabolism, but also plays a vital role in maintaining the stability of phospholipids in the cell membrane structure. Our previous transcriptome analysis of the YEND strain revealed that the expression levels of fatty acid degradation pathway-related genes, specifically *fadA* and *fadE* (encoding acetyl-CoA acyltransferase and acyl-CoA dehydrogenase, respectively), were significantly reduced by over eight times compared to the WT strain, suggesting a potential link between DksA and fatty acid metabolism in *Y. enterocolitica* [32]. In this study, we further verified the expression levels of related genes using RT-qPCR (see Appendix A for primers). As shown in Figure 1, the fatty acid synthesis pathway genes in YEND strains, such as *fabB* and *fabF*, were up-regulated by approximately two-fold compared with that of the WT strain, while the expression levels of genes involved in the fatty acid degradation pathway, including *fadA*, *fadD*, *fadE*, *fadI*, and *fadJ*, were down-regulated by more than five times. These results suggest that the DksA protein represses the synthesis of fatty acids in *Y. enterocolitica* while promoting the metabolism of fatty acids. 

### 2.2. DksA Played a Crucial Role in Facilitating the Expeditious Proliferation of Y. enterocolitica at the Optimal Temperature

*Y. enterocolitica* is a foodborne pathogen known for its ability to survive and propagate at low temperatures, indicating its high tolerance to cold environments [33]. However, the DksA protein has been found to induce the degradation of fatty acids, which is likely to alter the fluidity of cellular membranes and the sensitivity to varying temperatures, ultimately affecting the bacterial growth dynamics. To verify this, the growth conditions of the WT strain, YEND strain, and complementing strain YEND-D at 4 °C 16 °C, 26 °C, and 37 °C were measured, and the results are shown in Figure 2. At 4 °C, all strains had an OD_600_ value of approximately 0.3 after 24 h of growth, with no significant difference between the YEND and WT. However, as the temperature increased to 16 °C, 26 °C, and 37 °C, the growth rate of the YEND strain was consistently lower than that of the WT strain, while the YEND-D exhibited similar growth patterns to that of the WT strain at all tested temperatures. Therefore, this result confirmed the finding that DksA deletion would lead to deficient growth. Additionally, as the temperature increased, the growth gap between the WT and YEND strains gradually widened. As shown in Appendix A, at 21 h (approximately the stable growth period of the strain), the biomass of YEND was similar to that of WT at 4 °C, with only a slight growth gap of −3.31%. However, this growth gap significantly increased to 11.08% at 16 °C and further expanded to 15.08% and 23.95% at 26 °C and 37 °C, respectively. These findings suggest that DksA plays a pivotal role in facilitating the rapid growth process of *Y. enterocolitica* at optimal temperatures.

### 2.3. DksA Deletion Promoted Membrane Integrity in Y. enterocolitica

Fatty acids serve as essential building blocks for the synthesis of phospholipids and lipopolysaccharides found in the structure of the cell membrane structure. Disturbance in the metabolic pathways associated with fatty acids could alter the ratio of saturated and unsaturated fatty acids, thereby impacting the permeability of cell membranes [34]. To investigate the role of DksA in maintaining the integrity of the cell membrane structure, the permeability of the outer membrane and inner membrane of the WT strain, YEND strain, and complementing strain YEND-D were measured by NPN and ONPG assays.

NPN is a hydrophobic fluorescent probe capable of binding to hydrophobic structures that are exposed when the outer membrane of a cell experiences heightened permeability. The binding leads to an increase in the fluorescence signal of the probe, and the resulting fluorescence intensity reflects the changes in the bacterial outer membrane permeability. As shown in Figure 3, following a 10-min incubation in vitro, the YEND strain exhibited lower fluorescence values than the WT strain. In contrast, the fluorescence values of the YEND-D strain returned to levels similar to those of the WT strain. These results indicate that the YEND strain exhibits greater outer membrane permeability. In addition, we measured the stability of the inner membrane of *Y. enterocolitica*, and we evaluated the stability of the inner membrane of *Y. enterocolitica* using the ONPG. When the outer membrane of a cell is damaged, enzymes and nucleic acids can leak out, including β-galactosidase, which could hydrolyze ONPG to produce a yellow substance called o-nitrophenol. By quantifying the amount of color reaction, we can measure the leakage of extracellular enzymes and assess the stability of the bacterial inner membrane. Following a one-hour in vitro reaction, the absorbance value of the YEND strain was lower than that of the WT strain (Figure 3). These findings suggest that the inner membrane of the YEND strain was more stable and sustained less damage than the WT strain. Overall, the knockout of DksA resulted in greater integrity of both the outer and inner membranes of *Y. enterocolitica*.

### 2.4. DksA Positively Regulated the Polymyxin B Resistance of Y. enterocolitica

Polymyxin B is a cationic antimicrobial peptide antibiotic that targets the negatively charged phosphate group on the cell membrane of Gram-negative bacteria. By inserting into the fatty chain of the cell membrane, it destroys the structure of the cell membrane and increases the permeability of the cell membrane, leading to the leakage of small molecules such as purines and nucleotides [35,36]. In our previous study, we found that the minimum inhibitory concentration (MIC) of polymyxin B was 1 μg/mL for the WT, YEND, and YEND-D strains [37]. However, due to the strong bactericidal effects against *Y. enterocolitica* of polymyxin B, it may not be apparent whether DksA plays a role in the bacteria’s response to the antibiotic. Therefore, we conducted a more comprehensive analysis by measuring the growth conditions of the WT, YEND, and YEND-D strains at 0 MIC (0 μg/mL), 1/8 MIC (0.125 μg/mL), 1/4 MIC (0.25 μg/mL), and 1/2 MIC (0.5 μg/mL) to gain a more comprehensive understanding of the strains’ response to polymyxin B (Figure 4 and Appendix A). At 30 h (the stationary phase), the growth gap between the YEND and WT was 15.88% in the absence of polymyxin B. However, the addition of 0.125 μg/mL of polymyxin B resulted in an increase of the growth gap to 16.15%, which further rose to 17.35% when the concentration was increased to 0.25 μg/mL. Notably, the highest growth difference of 20.35% was observed under 0.5 μg/mL of polymyxin B. These results suggest that the DksA protein positively regulates the polymyxin B resistance of *Y. enterocolitica*.

### 2.5. DksA Deletion Rendered Y. enterocolitica Prone to Intracellular Nucleic Acid and Protein Leakage When Exposed to Polymyxin B

Previous findings have demonstrated that YEND has greater membrane integrity than WT, but exhibits lower resistance to polymyxin B, which suggests that the antibiotic causes more damage to the YEND cell membrane. To confirm this, the leakage of nucleic acid and protein under 1 MIC (1 μg/mL), 2 MIC (2 μg/mL), and 4 MIC (4 μg/mL) of polymyxin B were measured. As shown in Figure 5 and Figure 6, the amount of extracellular protein and nucleic acid detected gradually increased as the processing time continued to prolong, with the YEND exhibiting higher protein leakage compared to the WT at the same concentration of polymyxin B, suggesting that cell membrane integrity of the YEND strain is lower after polymyxin B treatment. Additionally, the amount of extracellular protein and nucleic acid, as well as the gap of the leakage amount between the YEND and WT were gradually increased as the concentration of polymyxin B increased, indicating that the YEND strain was more sensitive to polymyxin B. The above results verify that polymyxin B could damage the cell membrane structure of *Y. enterocolitica*, leading to the leakage of intracellular protein and nucleic acid, and that DksA helps *Y. enterocolitica* cope with the bactericidal effect of polymyxin B.

### 2.6. DksA Deletion Reduced Inner and Outer Membrane Integrity of Y. enterocolitica in the Presence of Polymyxin B

To further investigate the effect of DksA under polymyxin B exposure, outer membrane and inner membrane permeabilization of *Y. enterocolitica* were evaluated under 0.5 MIC (0.5 μg/mL) and 1 MIC (1 μg/mL) polymyxin B. As shown in Figure 7A, the YEND strain consistently displayed higher fluorescence values than the WT strain when exposed to 1 MIC of polymyxin B. Notably, the fluorescence value of the WT strain was higher than that of the YEND strain at the 10th min, suggesting that DksA contributed to attenuating the impact of polymyxin B on outer membrane permeability in *Y. enterocolitica*. Furthermore, the trend of fluorescence values for the YEND and WT at 0.5 MIC of polymyxin B was consistent with that at 1 MIC. In addition, the inner membrane permeabilization of the WT, YEND, and YEND-D when exposed to 1 MIC (1 μg/mL), 2 MIC (2 μg/mL), and 4 MIC (4 μg/mL) of polymyxin B were also evaluated. As shown in Figure 7B, the degree of damage to the inner membrane of *Y. enterocolitica* increased gradually with higher concentrations of polymyxin B and extended exposure durations. After approximately 5 h of treatment, inner damage was nearly complete, with the OD_410_ values of the YEND reaching 0.24 under 1 MIC polymyxin B, representing a 29.73% increase compared to the WT strain. Additionally, OD_410_ values of the YEND strain under 4 MIC polymyxin B were 102.44% higher than that of the WT strain, indicating that high concentrations of polymyxin B had a more significant impact on the YEND. These observations indicate that DksA plays a positive role in mitigating the harmful effects of polymyxin B on membrane damage in *Y. enterocolitica*.

## 3. Discussions

DksA is a transcriptional regulator that directly binds to RNAP secondary channels and plays an indispensable role in stringent response. It is generally believed that DksA functions to amplify (p)ppGpp signals during transcriptional regulation. (p)ppGpp inhibits the rRNA promoter by approximately three- to four-fold through the β′-ω interface site, whereas DksA binding to the secondary channel of RNAP can result in a twenty-fold inhibition [18]. In addition, DksA has been found to act in transcriptional regulation independently of (p)ppGpp. In *Haemophilus ducreyi* and *Y. enterocolitis*, DksA has been shown to govern approximately 17% and 22% of the entire open reading frames, respectively, involving transcription, macromolecular synthesis, energy metabolism, amino acid transport and utilization [38]. In our previous study, it was shown that the expression levels of several fatty acid metabolism genes, including *fadA*, *fadE*, and *fadI*, significantly decreased in DksA-deficient strains, suggesting a potential regulatory role of DksA in fatty acid metabolism and cell membrane integrity [32]. These results raise intriguing questions regarding the effect of DksA proteins on fatty acid metabolism in *Y. enterocolitis* and may have practical implications for the development of disinfectants targeting DksA in the food industry.

Here, it was confirmed that fatty acid metabolic genes were regulated by DksA using RT-qPCR. There was an approximately twofold decrease in the expression of fatty acid syntheses genes, such as *fabB*, *fabF*, and *fabR*, while several fatty acid degradation genes, including *fadA*, *fadD*, *fadE*, *fadI,* and *fadJ*, showed a more than fourfold decrease, with *fadA* exhibiting a significant ninefold decrease. Previous studies have extensively explored the effect of (p)ppGpp on fatty acid metabolism. Specifically, during the amino acid starvation, (p)ppGpp readjusts metabolic flux and regulates the expression of central metabolic genes, leading to the redistribution of acetic acid or β-oxidation originally used for fatty acid synthesis to the synthesis of glutamate precursors pyruvate and α-ketoglutarate [39]. In addition, fatty acid starvation induces the production of uncharged ACP, thereby inducing SpoT to synthesize a large amount of (p)ppGpp and turn on the stringent response [40,41]. Nevertheless, in *Y. enterocolitica*, only eight genes of fatty acid metabolism were significantly regulated (FC > 2), and KEGG analysis did not reveal significant enrichment. It is thus presumed that DksA functions independently from (p)ppGpp in regulating fatty acid metabolism pathways. FadR is a transcriptional regulator that controls the expression of fatty acid synthesis genes, such as *fabABHDG*. However, there was no significant change in the expression level of FadR in the YEND, and only a slight change in the relative expression level of FabABDG, suggesting that its regulation of fatty acid metabolism is not achieved through FadR, which requires additional research for detailed clarification.

The degradation pathway of fatty acids is not only a significant source of energy but also generates an essential intermediate product, acetyl-CoA, which functions as a critical intermediary in the metabolic process [42]. The YEND strain was found to have a growth defect, possibly due to insufficient energy and material caused by the downregulation of the fatty acid degradation genes. However, it is important to note that DksA, as a global regulator, influences bacterial life activities in various ways. Therefore, the growth defect may arise from multiple factors. Our previous study has also revealed that DksA positively regulates amino acid transport and utilization pathways [37]. Fatty acids constitute a primary component of phospholipids in the cell membrane structure, and the metabolism of fatty acids is closely related to the fluidity of cell membranes [43]. In this study, it was found that the YEND exhibits better growth in a low-temperature environment, and the growth gap between the WT strain and YEND strain gradually increased with temperature rise, indicating that the DksA is imperative for the rapid growth at optimal temperature. In *Shewanella baltica*, fatty acid degradation proteins were shown to positively regulate membrane fluidity, while fatty acid synthesis proteins negatively regulate it [44]. Given the reduced expression level of fatty acid degradation genes and increased expression level of fatty acid synthesis genes in the YEND strain, it is speculated that its cell membrane fluidity is inferior to that of the WT strain. And the negative impact of low temperature on the fluidity of the YEND cell membrane becomes limited. Additionally, the knockout of DksA would also lead to the upregulation of ribosomal proteins *rpaA*, *rplK*, *rpmA*, *rpmG*, and *rpmH*, as well as the transcription factors IF-1 and EF-Ts, which have been reported to promote cold adaptation of bacteria. All these reasons ultimately contribute to the insensitivity of the YEND strain to low temperature environments.

The periplasmic space of Gram-negative bacteria separates two layers of lipid membranes. The outer membrane (OM) consists of an inner leaflet mainly containing glycerophospholipids and an outer leaflet mainly containing lipid A, and its structural integrity directly affects the infectivity of bacteria. The inner membrane (IM) is composed of a phospholipid bilayer membrane and is used to isolate cellular metabolism and biosynthesis processes. Most proteins involved in electron transfer, protein export, and signal transduction are present in the IM, ensuring the basic life activities of bacteria [45,46]. It was shown that integrity of the inner and outer membranes of the YEND strain was higher than that of the WT strain, resulting in lower protein leakage. The specific impact of changes in fatty acid metabolism pathways on cell membrane structures, and whether there are other regulatory factors for DksA to affect cell membrane structure, are deed necessary to understand the roles of DksA in membrane integrity.

Polymyxin B is a cationic antimicrobial peptide antibiotic that has the ability to target the cell membrane and lyse both the OM and IM to release macromolecules, such as nucleic acid and cellular proteins, resulting in bacterial death. Previous studies have revealed that the WT strain and DksA-deficient strain are highly susceptible to polymyxin B, which can inhibit the growth of *Y. entriocolitica* at a concentration of 1 μg/mL. This study has demonstrated that the growth gap at the stationary phase decreases as the polymyxin B concentration increases, indicating that the DksA protein positively regulates the ability of *Y. entriocolitica* to resist polymyxin B. In addition, measurements of the leakage of cellular nucleic acid and protein in response to polymyxin B treatment indicate that the YEND is more severely lysed, whereas DksA can maintain the stability of the cell membrane under polymyxin B treatment. The integrity of the inner membrane and outer membrane also confirmed this. High concentrations of polymyxin B can cause damage to the inner and outer membranes of *Y. enterocolitica*, with the degree of damage being higher in the YEND strain compared to the WT strain. Since the YEND has higher cell membrane integrity in the absence of polymyxin B, it is suggested that DksA plays a positive role in response to the bactericidal effect of polymyxin B. The impact of the DksA protein on polymyxin B resistance may also be indirect. Our previous transcriptomic analysis revealed that DksA knockout results in a more than twofold decrease in phoP gene expression. Additionally, comparative analysis conducted by Guo et al. between the wild-type strain and the phoPQ-deletion strain demonstrated that the two-component system PhoP/Q enhances lipopolysaccharide synthesis and various lipopolysaccharide modification processes, thereby strengthening bacterial tolerance to polymyxin B [47]. Therefore, it is plausible that DksA indirectly enhances lipopolysaccharide synthesis through the PhoP/Q system, thus improving resistance to polymyxin B. However, further investigations are required to establish the precise connections.

## 4. Materials and Methods

### 4.1. Bacterial Strains and Growth Conditions

*Y. enterocolitica* ATCC23715 was defined as the wide type (WT), and the DksA gene was knocked out on this basis; the obtained strain was YEND. Then the pBAD24-DksA plasmid was transformed into the YEND strain by electroporation to obtain the complemented strain YEND-D [37]. The *Y. enterocolitica* strains were cultured in Luria–Bertani broth without salt (defined as LBNS) at 26 °C. The *E. coli* strains were routinely maintained in Luria–Bertani broth at 37 °C. When appropriate, 100 μg/mL of ampicillin or 50 μg/mL of kanamycin or CIN (15 μg/mL of cefsulodin, 4 μg/mL of irgasan, and 2.5 μg/mL of novobiocin) was added to the growth medium. In addition, 0.02 g/L L-arabinose was added to induce the expression of the DksA protein when the complemented strain was cultured.

### 4.2. Real-Time Quantitative PCR

RNA extraction methods, real-time PCR methods, and data analysis were performed as previously described [48]. *Y. enterocolitica* strains were cultured in an LBNS medium at 26 °C to the mid-logarithmic phase and the total RNA of strains was extracted using an RNAprep pure Cell/Bacteria Kit (Tiangen, Beijing, China) according to the manufacturer’s instructions. The integrity of the extracted RNA was determined by agarose electrophoresis, and the concentration was determined using a Nanodrop 2000c. A two-step method was used to perform RT-qPCR experiments. First, 200 ng of total RNA was reverse-transcribed with PrimeScriptTM RT Master Mix (Takara, Kusatsu, Japan) to obtain cDNA, and then RT-qPCR was performed using the TB Green^®^ Premix Ex Taq™ II (Takara) kit. The complete list of primers used in this study can be found in Appendix A. 16sRNA was used as a reference for normalization.

### 4.3. Growth Conditions

Two methods were used to determine the growth curves in this study. To analyze the growth condition of *Y. enterocolitica* with different concentrations of polymyxin B, a fresh overnight bacterial culture was diluted with LBNS to an OD_600_ of 0.04, and then 100 μL of diluted culture was added to each well of the 100-well plate with 100 μL of a certain concentration of polymyxin B liquid, and then 100 μL of diluted culture was added to each well of the 100-well plate with 100 μL of a certain concentration of polymyxin B prepared with an LBNS liquid medium. The growth of *Y. enterocolitica* was cultured at 26 °C and monitored at 600 nm using an automatic curve growth instrument every hour. The experiment was performed in at least two independent trials with five biological replicates. For the shake flask method, a fresh overnight bacterial culture was diluted into a 250 mL shake flask containing 50 mL of a fresh LBNS medium at an OD_600_ of 0.02 and cultured at 180 rpm with specific temperature. Then, 200 μL samples of each strain were taken and measured at OD_600_ using a microplate spectrophotometer (Puxi University, Co., Ltd., Beijing, China). This experiment was performed in at least two independent trials with three biological replicates.

### 4.4. Outer Membrane Permeabilization

N-phenyl-l-naphthylamine (NPN) uptake assays were used to determine the outer membrane permeability of *Y. enterocolitica*, as previously described [49]. Specifically, the fresh cultivated *Y. enterocolitica* liquid culture was inoculated into a fresh 100 mL LBNS medium according to the inoculum volume of 1%, and an additional 0.02 g/L of arabinose was added for the complementing strain, to mid-logarithmic phase. The cells were harvested by centrifuging and washed with a sterile phosphate buffer saline (PBS) buffer three times. A 1.5 mL cell resuspension of OD_600_~0.5, required polymyxin B, and a 10 μM working concentration of NPN fluorescent staining were both added into an EP tube. Then we took 200 μL of the mixture in a 96-well fluorescent plate, adjusted the excitation wavelength to 350 nm and the absorption wavelength to 420 nm, and collected data. The experiment was performed in five biological parallels at least twice.

### 4.5. Inner Membrane Permeabilization

O-nitrophenyl-*β*-D-galactoside (ONPG) assays were performed to evaluate inner membrane permeability as previously described [49]. Briefly, a fresh overnight bacterial culture was inoculated into 100 mL of LBNS medium at 1% and then cultured to the mid-logarithmic growth phase. The cells were harvested by centrifugation at 5000 rpm, washed with PBS buffer twice, and then resuspended to OD_600_ at 0.5. 1.5 mL of bacterial solution, and required polymyxin B, as well as an ONPG working concentration of 30 mM mixed in a tube. Then, 200 μL of the mixture was added into each well of a 96-well plate, and the absorbed data were measured at 410 nm using a spectrophotometer. The experiment was performed in five biological parallels at least twice.

### 4.6. Protein and Nucleic Acid Leakage Assays

The fresh cultivated *Y. enterocolitica* liquid was inoculated into 100 mL of the LBNS medium and cultured to the mid-logarithmic growth phase. The cells were then washed with PBS buffer twice and resuspended to OD_600_ at 0.5. 10 mL of the bacterial solution, and polymyxin B as required, were fully mixed in an EP tube. Then, 1 mL of the mixture was taken every hour and centrifuged at 12,000 rpm. The supernatant was used to determine the protein and nucleic acid leakage amount using a Nano-drop 2000 [50]. The experiment was performed in three biological parallels at least three times.

### 4.7. Statistical Analysis

Statistical analysis was performed using a one-way analysis of variance with Dunnett’s multiple-comparison test. *** *p* < 0.001. 

## 5. Conclusions

In summary, this study identified that DksA positively regulated the fatty acid degradation pathway and negatively regulated the fatty acid synthesis pathway. In addition, the absence of the DksA protein enhances the integrity of both the inner and outer membranes in *Y. enterocolitica*, which may explain the insensitivity of the YEND to low temperature environments. And DksA would promote the integrity of the IM and OM of *Y. entriocolitica* when treated with polymyxin B, thereby eventually leading to a reduced sensitivity to polymyxin B. These findings have broadened our knowledge on the role of DksA in regulating fatty acid metabolism and membrane structure, and have provided insights into the bactericidal mechanism of polymyxin B.

## Figures and Tables

**Figure 1 ijms-24-11951-f001:**
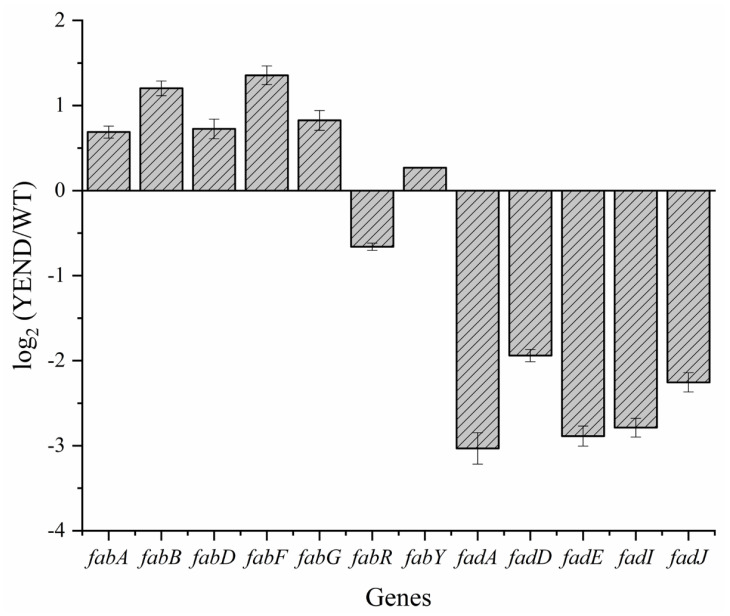
Transcriptional changes of fatty acid metabolic genes of YEND. Total RNAs were extracted from mid-log phase *Y. enterocolitica*. The RNA-seq data is from our previous study, and the difference of RNA-seq data indicated the changes in fatty acid metabolic genes between YEND and WT strains (normalized to 16sRNA). Data are means from three independent RT-qPCRs.

**Figure 2 ijms-24-11951-f002:**
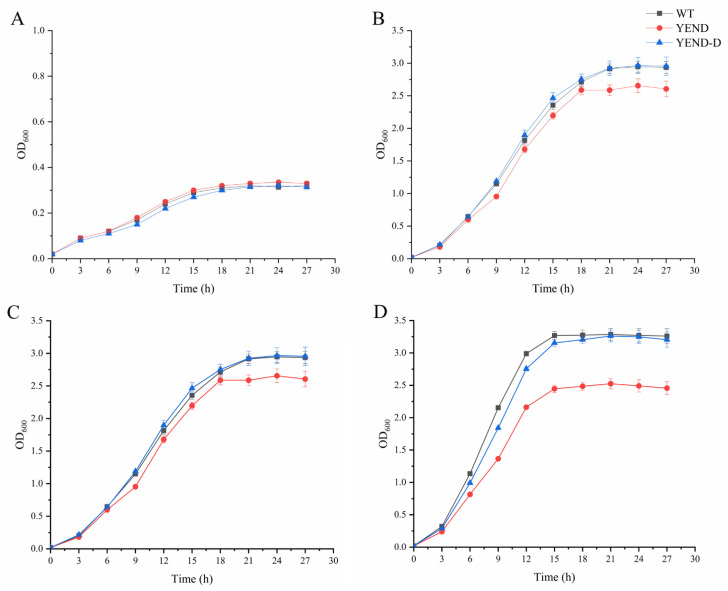
Growth conditions of *Y. enterocolitica* at 4 °C (**A**), 16 °C (**B**), 26 °C (**C**), 37 °C (**D**) supplemented with 0.02% L-arabinose. Data are mean OD_600_ values for three independent experiments and standard errors of the means.

**Figure 3 ijms-24-11951-f003:**
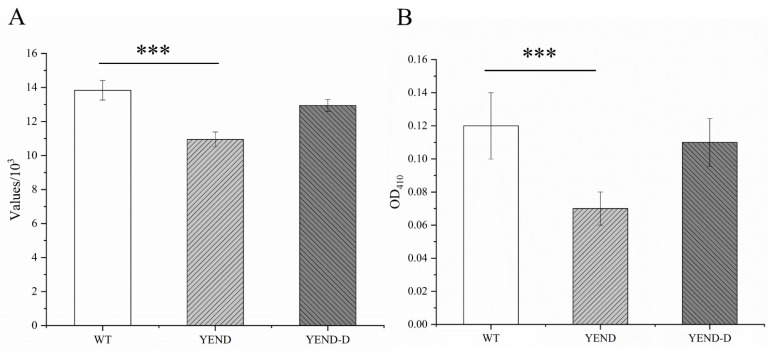
Effect of DksA on *Y. enterocolitica* cell membrane permeability. Outer (**A**) and inner (**B**) membrane permeability was evaluated by NPN and ONPG, respectively. The experiment was performed at least three times with five biological parallels. The data are presented as the mean ± SD of at least three independent experiments. Asterisks indicate a significant difference. ***, *p* < 0.001.

**Figure 4 ijms-24-11951-f004:**
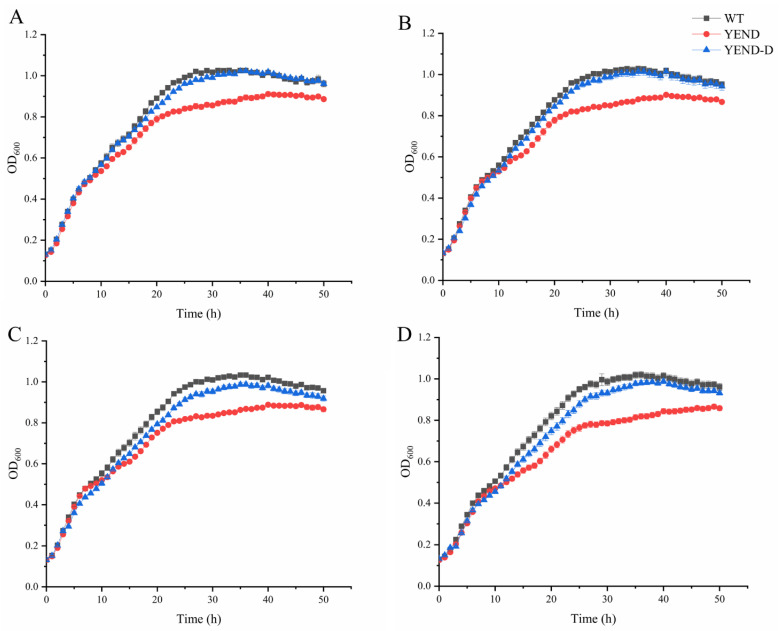
Growth conditions of *Y. enterocolitica* exposed to 0 μg/mL (**A**), 0.125 μg/mL (**B**), 0.25 μg/mL (**C**), and 0.5 μg/mL (**D**) polymyxin B supplemented with 0.02% L-arabinose. Data are mean OD_600_ for five independent cultures and standard errors of the means.

**Figure 5 ijms-24-11951-f005:**
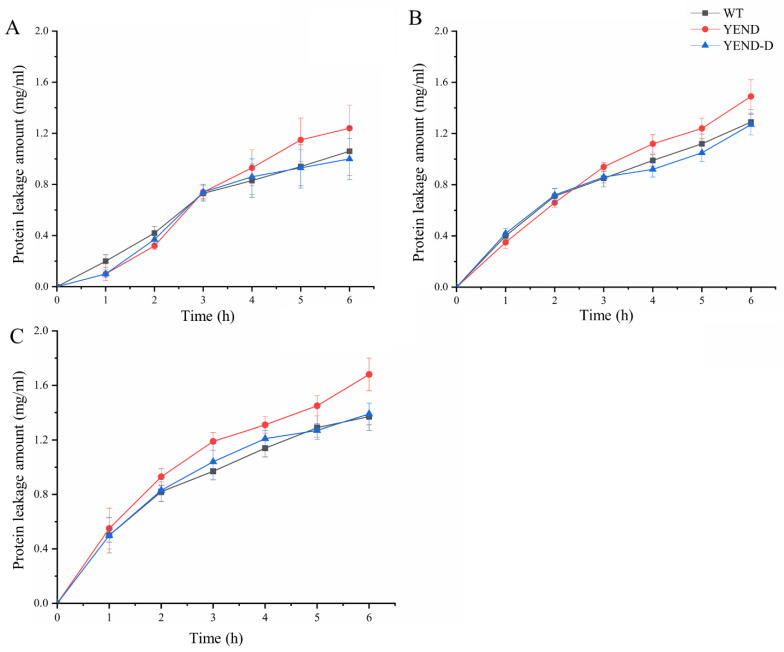
Protein leakage of *Y. enterocolitica* strains exposed to 1 μg/mL (**A**), 2 μg/mL (**B**), 4 μg/mL (**C**) polymyxin B supplemented with 0.02% L-arabinose. The experiment was performed at least three times with three biological parallels. The data are presented as the mean ± SD of at least three independent experiments.

**Figure 6 ijms-24-11951-f006:**
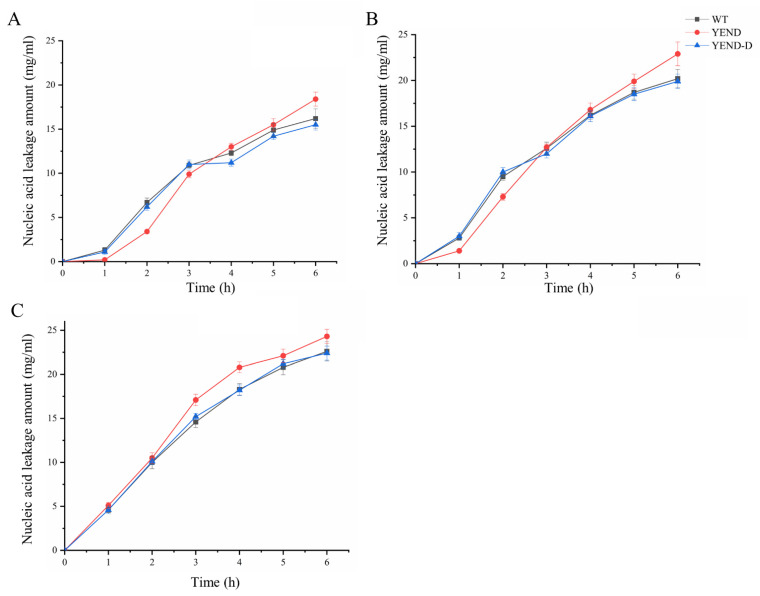
Nucleic acid leakage of *Y. enterocolitica* strains exposed to 1 μg/mL (**A**), 2 μg/mL (**B**), 4 μg/mL (**C**) polymyxin B supplemented with 0.02% L-arabinose. The experiment was performed at least three times with three biological parallels. The data are presented as the mean ± SD of at least three independent experiments.

**Figure 7 ijms-24-11951-f007:**
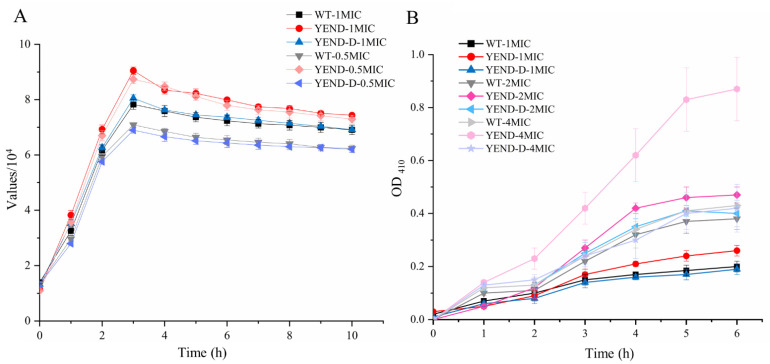
Outer and inner membrane permeability of *Y. enterocolitica* exposed to polymyxin B. (**A**) Outer membrane permeability exposed to1 μg/mL (1 MIC) and 0.5 μg/mL (0.5 MIC) polymyxin B was evaluated by NPN. (**B**) Inner membrane permeability exposed to1 μg/mL (1 MIC), 2 μg/mL (2 MIC), and 4 μg/mL (4 MIC) polymyxin B was evaluated by ONPG. The data are presented as the mean ± SD of at least three independent experiments.

## Data Availability

The remaining research data are available in the Appendix A.

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
