# Peer review of "Stringent Response Factor DksA Contributes to Fatty Acid Degradation Function to Influence Cell Membrane Stability and Polymyxin B Resistance of Yersinia enterocolitica"

_ijms, 2023, doi:10.3390/ijms241511951_

Round 1

Reviewer 1 Report

In this paper (“Stringent response factor DksA contributes to fatty acid degradation function to influence cell membrane stability and polymyxin B resistance of Yersinia enterocolitica”), Huang, et al. investigate the role of DksA in regulating the transcription of genes involved in fatty acid metabolism. They find the expression of genes in fatty acid degradation and biosynthesis are altered by deletion of dksA and that the mutant has lower envelope permeability than the wild type; however, with polymyxin treatment the mutant showed increased envelope damage. This is an interesting and well written paper. However, several there are several points in the writing and experimentally that should be addressed to strengthen the paper.

Specific comments:

1.     Well clearly understandable, the paper would be improved by editing for verb tenses (to match singular vs. plural verbs) and some typos.

2.     Table 1: Description of the BL21 (dksA) strain is not correct.

3.     Fig. S1: This figure needs a positive control of a promoter that DksA is known to bind to in order to show that the assay is working.

4.     Has the role of FadR in the regulation observed been investigated?

5.     Ln 233: The framing of this paragraph is a little confusing as it is suggested that DksA would be important for grow at low temperatures and than as that this growth effect is confirmed because the dskA mutant has a growth defect at high temperatures. This wording should be adjusted.

6.     Ln 259: It should be mentioned here that fatty acids are also necessary for LPS synthesis and that changes in the saturation state of fatty acids synthesized can also change LPS/phospholipid ratios as LPS has only saturated fatty acids.

7.     Ln 266-273: The framing of the NPN as measuring “OM damage” and the decreased permeability of the dskAmutant as less damaged is not entirely accurate. The assay is an assay of OM permeability and the mutant is less permeable than the wild type. This does not mean the OM of the wild type is “damaged”.

8.     Ln 273-274: A sentence is repeated.

9.     The altered envelope permeability and increased susceptibility to polymyxin observed in the dskA mutant would be consistent with increased LPS production. The levels of LPS in this strain should be assayed. At the least, the possibility of altered LPS synthesis should be included in the discussion.

The paper is well written; however, there are grammatical and typing errors that should be addressed.

Reviewer 2 Report

Manuscript deals with characterization of the function of DksA on fatty acid metabolism and cell membrane structure in Yersinia enterocolitica just by analysis of gene expression, growth conditions, and membrane permeabilization. By using RT-qPCR, it shows that fatty acid synthesis pathway genes were up-regulated by approximately two-fold in the deletion mutant compared with that of the WT strain, while the expression levels of genes involved in fatty acid degradation pathway were down regulated by more than five times. By EMSA the increase in the molar ratio of DksA protein did not significantly block DNA fragment, indicating that fadA, fadD, fadE, fadI and fadJ were not directly regulated by DksA

These results are well defined.

line 251-252 "These findings suggest that DksA is dispensable for the growth of Y. enterocolitica at low temperatures but plays a pivotal role during the rapid growth of the pathogen" This conclusion is not correct, because what it is shown in growth curves is an increasing dependance of DskA as a function of temperature. So, the effect at low temperature is lower. Nevertheless, this experiment is not very informative about the functionality of DksA.

The rest of experiments trying to identify some relevant effect of DksA on cell growth and membrane integrity are very indirect and do not provide any relevant data. Even if the experiment was performed at least three times with five biological parallels.

Why only one of the biological parallels was chosen to get the mean and standard deviation?

The manuscript cannot be published in the actual form, and only if some more relevant experiments are going to be done, it could be consider; or just a note about the DksA regulatory role should be published .

Minor editing of English language required

Round 2

Reviewer 1 Report

In this paper (“Stringent response factor DksA contributes to fatty acid degradation function to influence cell membrane stability and polymyxin B resistance of Yersinia enterocolitica”), Huang, et al. investigate the role of DksA in regulating the transcription of genes involved in fatty acid metabolism. They find the expression of genes in fatty acid degradation and biosynthesis are altered by deletion of dksA and that the mutant has lower envelope permeability than the wild type; however, with polymyxin treatment the mutant showed increased envelope damage. This is an interesting and well written paper. The revisions to the paper in the current version have addressed all previous concerns.

Reviewer 2 Report

The requested revision has been addressed correctly.